# Effect of Coiling Temperature on Microstructures and Precipitates in High-Strength Low-Alloy Pipeline Steel after Heavy Reduction during a Six-Pass Rolling Thermo-Mechanical Controlled Process

Yicong Lei [1], Wen Yang [1], Charles W. Siyasiya [2] and Zhenghua Tang [1,*]

1. College of Materials Science and Engineering, Sichuan University, Chengdu 610065, China; 2022323015027@stu.scu.edu.cn (Y.L.); 2021223010047@stu.scu.edu.cn (W.Y.)
2. Department Materials Science & Metallurgical Engineering, University of Pretoria, Pretoria 0002, South Africa; charles.siyasiya@up.ac.za
* Correspondence: sacdtzh@163.com; Tel.: +86-138-0802-1804

**Abstract:** Nb-Ti high-strength low-alloy pipeline steel was subjected to a six-pass rolling process followed by the coiling process at different temperatures between 600 and 650 °C using the thermo-mechanical testing system Gleeble 3500 (Gleeble, New York, NY, USA). This experimental steel was subjected to 72% heavy reduction through a thermos-mechanical controlled process. Thereafter, the microstructures were observed using optical microscopy, scanning electron microscopy, electron backscatter scanning diffraction, and transmission electron microscopy coupled with energy dispersive spectrometry and selected area electron diffraction. For the selected three coiling temperatures of 600, 625, and 650 °C, acicular ferrite, polygonal ferrite, and pearlite were observed, and morphology and statistical analysis were adopted for the study of precipitates. Based on the estimation by the Ashby–Orowan formula, the incremental strength through precipitation strengthening decreases with coiling temperatures and reaches 26.67 Mpa at a coiling temperature of 600 °C. Precipitation-time-temperature curves were obtained to explain the transformation of precipitates. The (Nb, Ti)(C, N) particles tended to precipitate in the acicular ferrite with $[011]_{(Nb, Ti)(C, N)}//[011]_{\alpha\text{-Fe}}$ orientation. The lower coiling temperature provided enough driving force for the nucleation of precipitates while inhibiting their growth.

**Keywords:** coiling temperature; multi-pass; TMCP; heavy reduction; steel

## 1. Introduction

High-strength low-alloy (HSLA) steel is regarded as a series of low-carbon steel micro-alloyed with V, Nb, or Ti, with a yield strength reaching 690 Mpa [1–4]. HSLA steels are extensively utilized for several engineering applications due to their high yield strength and cold formability [5–7]. For example, HSLA steel is used in automobile components, petroleum pipelines, load-bearing parts of cranes, submarine constructions, ships, bridges, railways, and so on. Most HSLA steel can be manufactured through the thermos-mechanical controlled process (TMCP) [8,9]. However, an increase in strength is often accompanied by a decrease in fracture toughness [10]. Therefore, there is a need to develop materials with a combination of high strength and toughness to meet the requirements of safety and practicality.

Based on these requirements, certain heat treatment processes (such as quenched and tempered [11]) and transformation-induced plasticity provide suitable HSLA steels with high strength and toughness [12–15]. As an effective method, TMCP has been extensively employed to produce HSLA steel [16]. The microstructures and mechanical properties can be manipulated by processing parameters, such as pass strain, rolling temperature, cooling rate, coiling temperature (CT), etc. [11,16–18]. In particular, CT is known to have

a strong effect on nano-scale precipitates [19–23]. For Nb-Ti HSLA pipeline steel, Nb and Ti play a significant role in achieving their strength [3,24], as the precipitation of Nb and Ti is very sensitive to CT [19,20]. For Nb-Ti micro-alloyed steel, the precipitates begin to form at a CT of 675 °C. The intermediate precipitates are small in size and randomly distributed in the matrix at a CT of 650 °C [20–25]. During the coiling process in TMCP, large (Nb, Ti)C particles and small spherical (Nb, Ti)C particles form in Nb-Ti HSLA pipeline steel, which leads to dislocation pinning [21]. It is generally believed that microstructural factors contribute to mechanical properties, such as grain refinement and precipitation hardening [26–28]. Liu Y.J. et al. [29] studied the effect of coiling temperature on the microstructure and properties of a Ti-Nb micro-alloyed steel with a two-pass rolling process. Patra P.K. et al. [30] compared the effect of two coiling temperatures below 580 °C in Nb-Ti steel. However, there are very few studies on how CT affects the microstructures and precipitation behavior in Nb-Ti HSLA pipeline steel during heavy reduction TMCP.

In this study, a six-pass rolling process was deployed in TMCP to obtain a smaller grain size and phase transformation driving force, which has practical engineering applications. Nb-Ti HSLA pipeline steel subjected to the TMCP process with heavy reduction was hot-rolled and then coiled at different temperatures. Microstructures, morphology, and statistical analysis of precipitates were analyzed to elucidate the effect of the CT. Furthermore, the dynamic precipitation behavior during the reduction process was analyzed using precipitation–time–temperature (PTT) diagrams.

## 2. Materials and Methods

The steel used in this experiment was smelted in a 25 kg vacuum induction furnace. The size of cast ingot was 43 mm × 66 mm × 235 mm and the sample was obtained near the tail area after removing the tail. The chemical composition of the steel was analyzed by a direct reading spectrometer (OBLF-QSN750, OBLF, Dortmund, Germany) and the results are shown in Table 1.

**Table 1.** The chemical composition of experimental steel (wt%).

| C | Si | Mn | S | P | Ti | Nb | V | N | Al |
|---|---|---|---|---|---|---|---|---|---|
| 0.093 | 0.376 | 1.540 | 0.005 | 0.0084 | 0.094 | 0.078 | 0.000 | 0.005 | 0.013 |

The as-cast blocks were homogenized at 1250 °C for 300 s and cylinder specimens (10 mm × 15 mm) were prepared for compression testing on the thermos-mechanical testing system Gleeble 3500® (Gleeble, New York, NY, USA). Figure 1 shows the schematic diagram of the hot compression set-up and Figure 1b shows the schematic diagram of the TMCP schedules showing a six-pass schedule, which was obtained from previous studies [23]. The strain rate of 1 s$^{-1}$ was kept during reduction. The first pass was applied at 1100 °C. The strain values of the first pass and the final pass were 0.25 and 0.15, respectively, and the other passes were 0.2. The final rolling temperature was 840 °C. Ultimately, these samples were subjected to the coiling process for 3600 s at 650, 625, and 600 °C, respectively, and thereafter air cooled down to room temperature. The finished height of the cylindrical sample was 4.20 mm after 72% heavy reduction.

The cross-sections of Gleeble® test samples were observed under an optical microscope (OM), a scanning electron microscope (SEM), and using electron backscatter scanning diffraction (EBSD). The cross-section of the sample was ground, polished, and finally electro-polished in a 14% perchloric acid + 86% alcohol solution for 20 s using a voltage of 25 V at room temperature. The EBSD was characterized by a field emission SEM (ZEISS Gemini 300, ZEISS, Oberkochen, Germany) with an operating voltage of 25 kV. The EBSD map was obtained In the range of 0.25–0.5 μm step size.

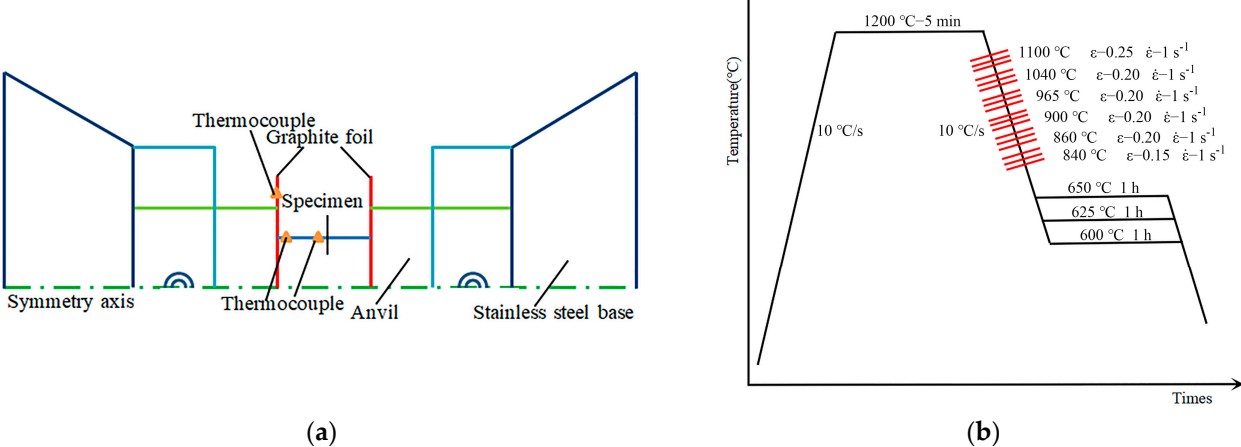

**Figure 1.** (**a**) Schematic diagram of hot compression in Gleeble 3500; (**b**) schematic diagram of the thermos-mechanical processing schedule involving 6-pass reduction. Abbreviations: $\dot{\varepsilon}$—strain rate; $\varepsilon$—strain.

Thin foils were prepared through the punching and ion-milling process and viewed through a Tecnai G2 F20 S-TWIN Transmission Electron Microscope (TEM, FEI, Hillsboro, OR, USA) at 200 kV. The microstructure and the precipitates of the samples were investigated under high-resolution transmission electron microscopy (HRTEM). The operating voltage of TEM was 300 kV. The TEM coupled with an energy dispersive spectrometer (EDS) was used to characterize the elemental composition of the precipitates.

Carbon replicas were prepared to observe the precipitation behavior of carbonitride. The polished sample was etched with 2% nital solution and then placed into a vacuum evaporation instrument to cover the surface of the sample with sprayed carbon. After spraying, the carbon layer was scratched into $4 \times 4$ mm$^2$ with a blade and soaked into the solution to separate the carbon film from the matrix. The carbon films were dried, loaded onto the copper grid, and observed using the JEM-2100F TEM (JEOL, Tokyo, Japan).

## 3. Results

### 3.1. Microstructures and CCT Diagrams

Figure 2 shows the CCT curves of experimental steel calculated by JMatpro 7.0 software. JMatpro assumed a grain size of 20 μm after austenitization at 900 °C. It can be seen from the figure that the ferrite and pearlite start temperatures were predicted to be 845 and 687 °C, respectively. As can be seen, the microstructures were predicted to be mainly ferrite and pearlite at a cooling rate of 10 °C/s.

Figure 3 shows the microstructure of the steel at different CTs of 600, 625, and 650 °C, respectively. Figure 3a–c show that the microstructure was mainly composed of acicular ferrite (AF), polygonal ferrite (PF), and pearlite (P) as labeled. After being etched by 2% nital solution, P could be seen as a dark area under OM. The volume fraction of the P was calculated by its area percentage of the OM micrographs. As can be seen in Figure 4, the volume fractions of P were found to be 11.48, 11.68, and 12.37% at CTs of 600, 625, and 650 °C, respectively.

AF is characterized by a fine non-equiaxed ferrite of parallel or interwoven non-parallel ferrite laths with various sizes. Besides confirming the OM observations, the SEM secondary electron images (SEIs) revealed that the AF clusters exhibited irregular shapes and different sizes. Normally, AF cannot grow across the prior austenite grain boundaries. Thus, individual ferrite cluster was transformed from individual austenite grains [31]. AF grains with different shapes and sizes were influenced by the austenite size distribution after the TMCP.

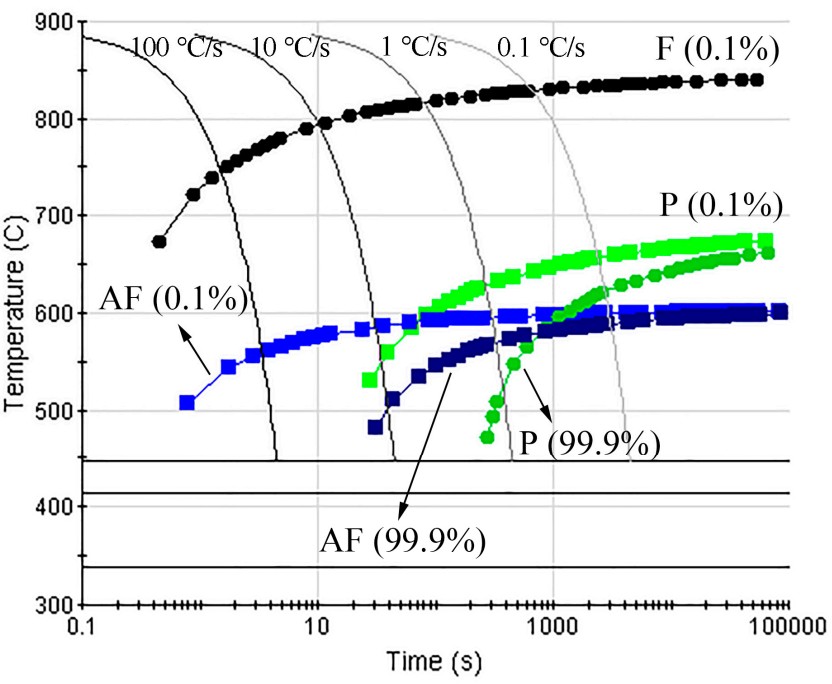

**Figure 2.** Continuous cooling transformation diagram of the experimental HSLA steel calculated using JMatPro (F for ferrite; AF for acicular ferrite; and P for pearlite; Black for F (0.1%); light green for P (0.1%); dark green for P (99.9%); light blue for AF (0.1%); dark blue for AF (99.9%)).

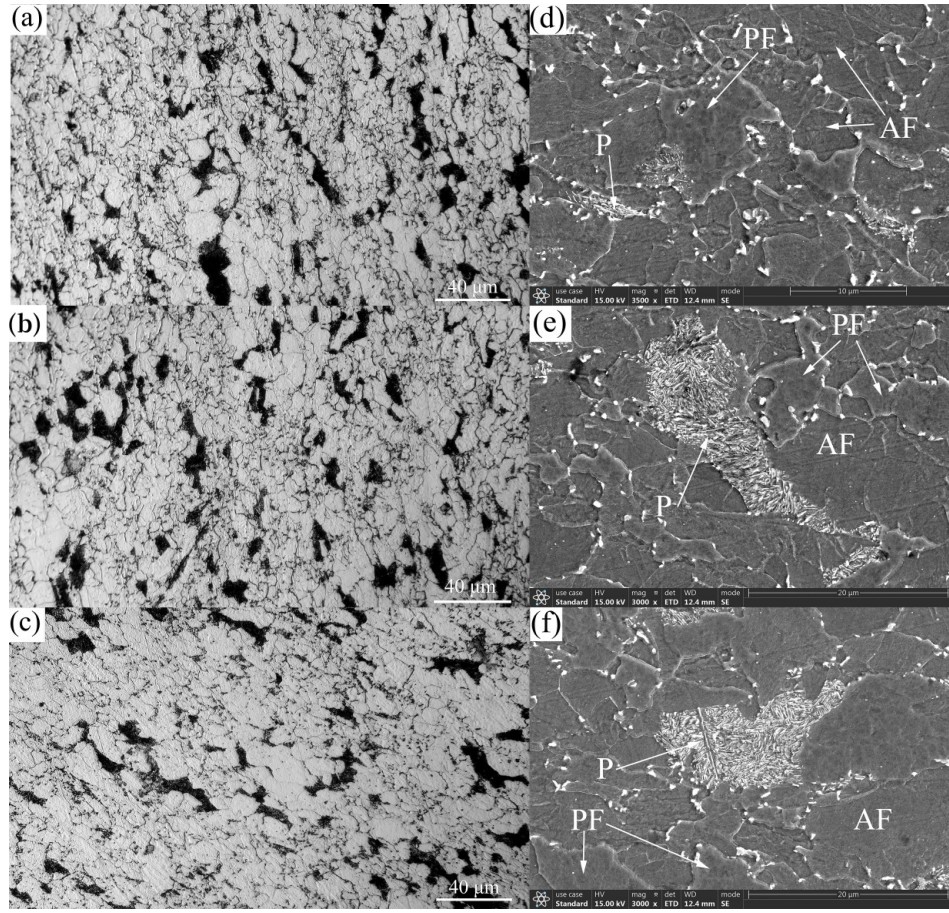

**Figure 3.** (**a**–**c**) Optical micrographs and (**d**–**f**) SEM micrographs of TMCP Nb-Ti HSLA pipeline steel at different coiling temperatures: (**a**,**d**) 600 °C; (**b**,**e**) 625 °C; and (**c**,**f**) 650 °C.

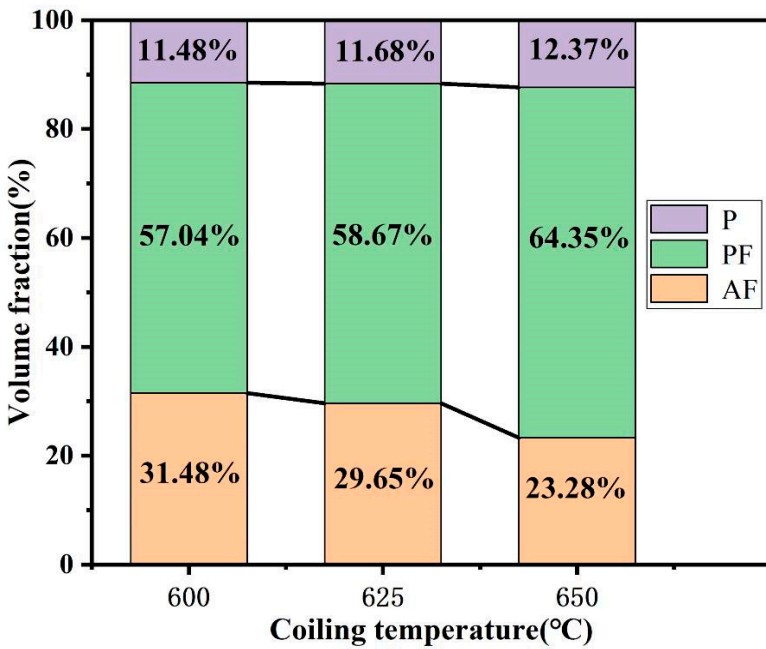

**Figure 4.** The volume fraction values of P, AF, and PF of the TMCP Nb-Ti HSLA pipeline steel after coiling at 600, 625, and 650 °C.

Figure 5 shows the inverse pole figure (IPF) of the longitudinal section of the deformed samples. The high-angle grain boundaries (HAGBs, blue) referred to grain misorientations larger than 15°, while the low-angle grain boundaries (LAGBs, green) referred to grain boundaries with grain misorientations less than 15°. The quantification of AF can be evaluated by the EBSD quantification standard. The AF generally has a large aspect ratio and a high dislocation density in micro-alloyed steel. In addition, AF has a high level of crystallographic orientation change compared with PF as shown in Figure 5. It has been reported that AF is associated with the LAGBs [32]. As shown in Figure 6a, a high proportion of LAGBs existed at a CT of 600 °C, which is probably related to the higher volume fraction of AF. Therefore, the volume fraction of the AF was calculated from the EBSD data and was found to be 31.48, 29.65, and 23.28% for CTs of 600, 625, and 650 °C, respectively. In other words, the volume fraction of the AF gradually decreased with the CT.

Figure 6b shows the grain size distribution for the three CTs. The calculation of average grain size and grain size distribution was based on the diameter of the equiaxed circular area. The grain size distribution provides a more robust method for the analysis of the grain size compared with the average grain size. The CT of 600 °C exhibited the largest fraction of grains with a size of less than 4 μm and the smallest average grain size of 4.42 μm. On the contrary, when the CT was raised to 650 °C, the grain size was found to be between 5 and 11 μm with an average grain size of 5.37 μm. In other words, the grain size significantly decreased with the CT.

Figure 7 shows the detailed morphology of AF observed under TEM. The AF grain size was found to be between 0.31 μm to 1.48 μm with high-density dislocations and dislocation cells. The PF showed almost less dislocation due to its diffusional phase transformation. It can be deduced that the size of PF grains was small and the microstructure was uniform at lower CTs. On the contrary, the AF exhibited a complex dislocation structure. Both the high density and a large number of dislocation cells were generated by the mutual entanglement of dislocations. This dislocation structure can substantially increase the stress required for the dislocation slip and increase the strength of the steel [33]. In general, there was higher dislocation density at lower CTs. These dislocations also served as effective nucleation sites for precipitates. In other words, the CT of 600 °C was optimal for a good combination of strength and toughness.

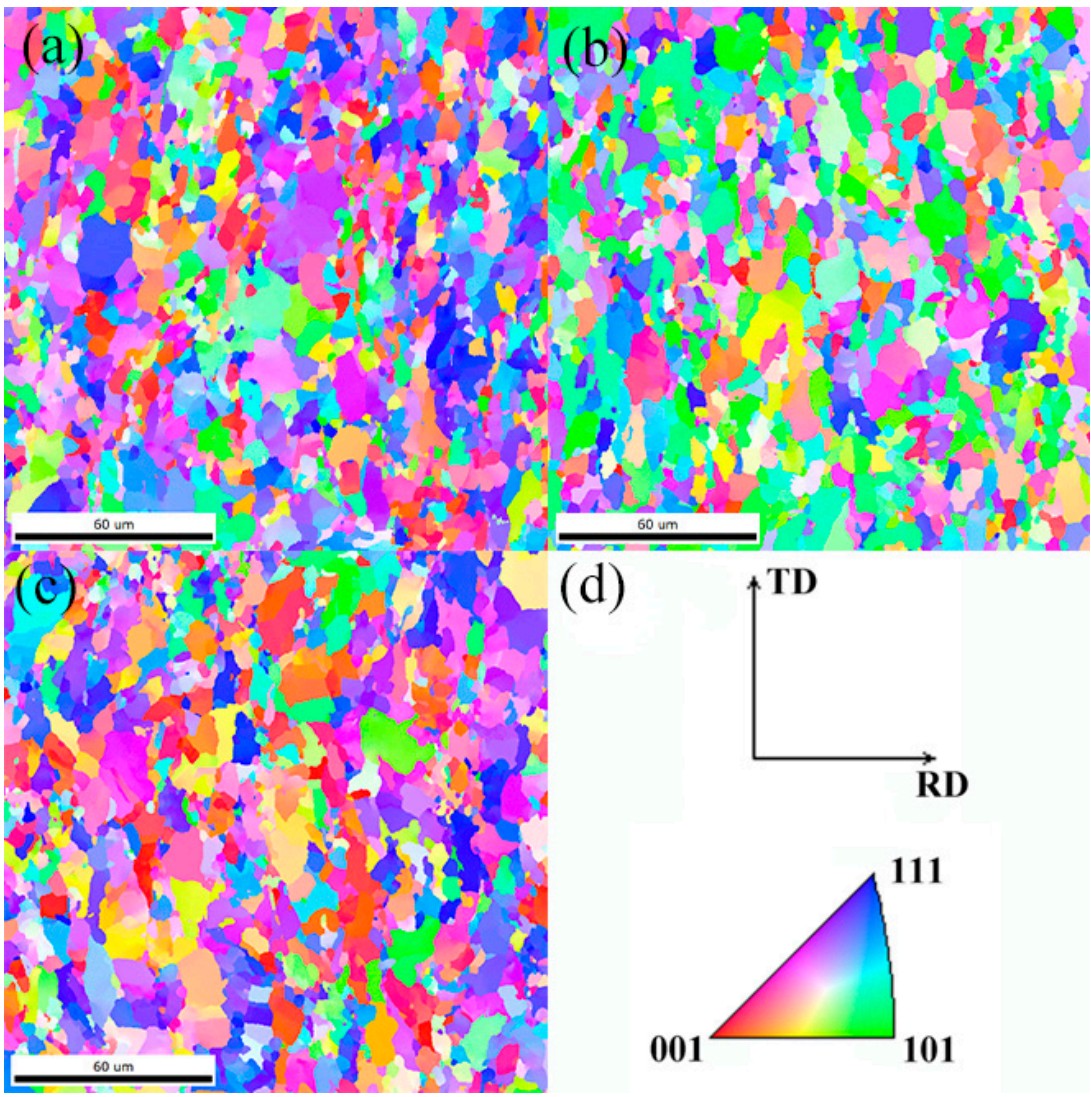

**Figure 5.** IPF maps taken from the longitudinal section at CTs of (**a**) 600 °C, (**b**) 625 °C, and (**c**) 650 °C. (**d**) Inverse pole figure color key.

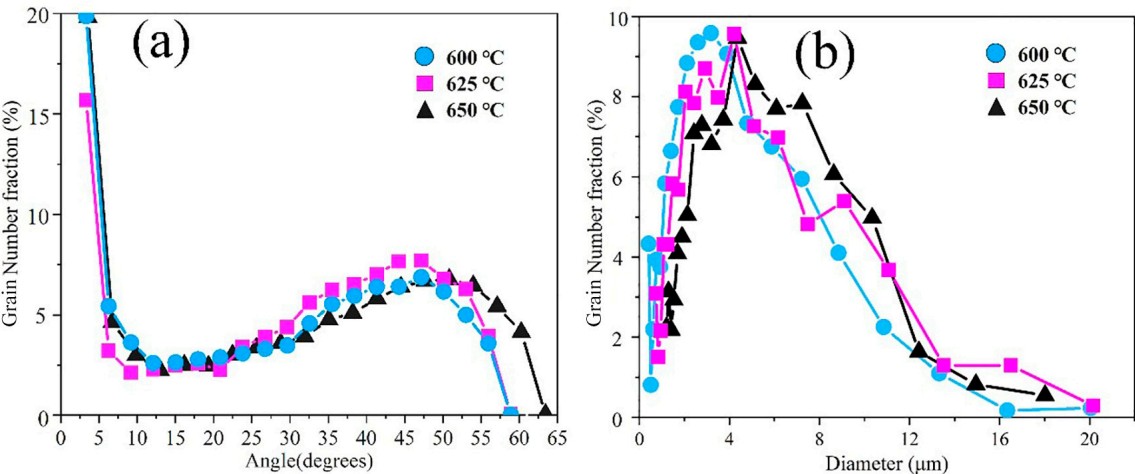

**Figure 6.** (**a**) Grain boundary angle distribution and (**b**) grain size distribution at CTs of 600, 625, and 650 °C.

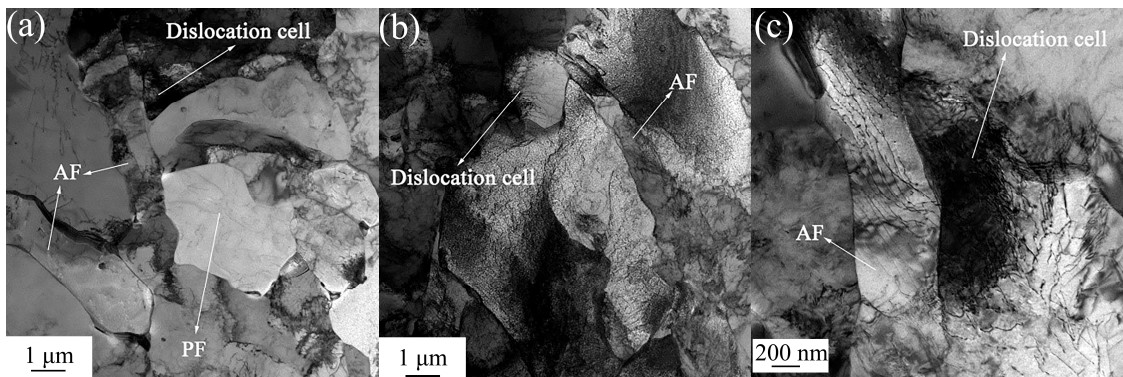

**Figure 7.** The AF morphology of TMCP Nb-Ti HSLA pipeline steel observed under TEM at different coiling temperatures of (**a**) 600 °C, (**b**) 625 °C, and (**c**) 650 °C.

Figure 8 illustrates the P microstructures observed under TEM. The interlamellar spacing was found to be 129.5, 148.5, and 150.4 nm at CTs of 600, 625, and 650 °C, respectively, i.e., the lamellar spacing increased with the CT. The carbonitrides that were induced by the reduction formed near the austenite grain boundaries. The reduction of carbon content in austenite is conducive to P transformation [34].

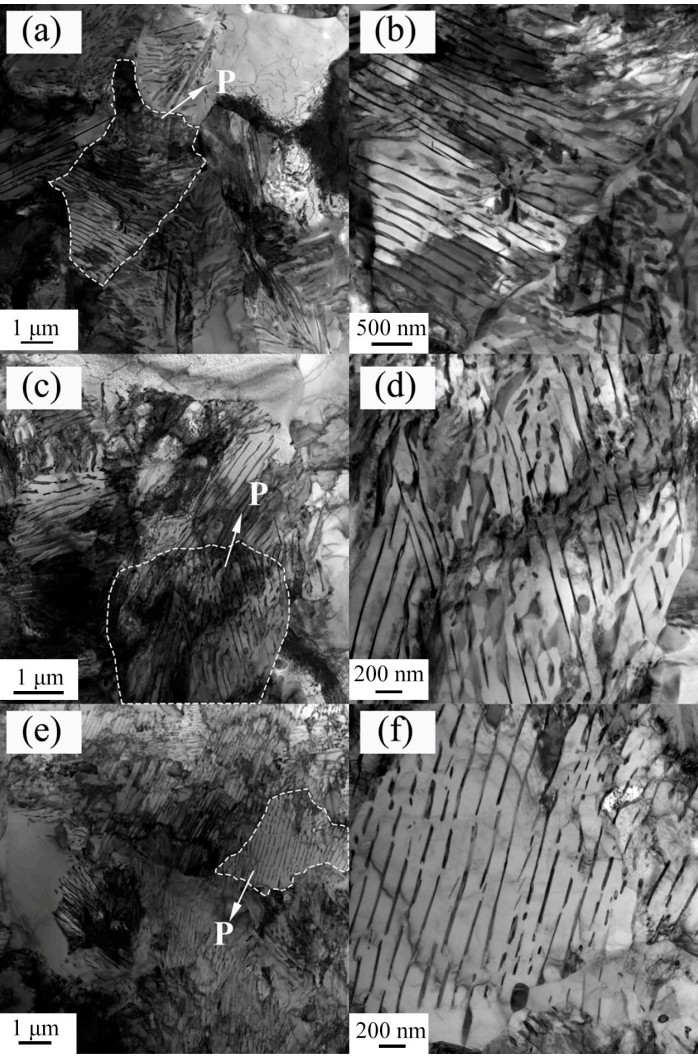

**Figure 8.** The morphology of the pearlite observed under TEM at CTs of (**a**) and (**b**) 600 °C; (**c**) and (**d**) 625 °C; and (**e**) and (**f**) 650 °C.

### 3.2. Morphology and Statistical Analysis of Precipitates

Figure 9 shows a thin foil TEM micrograph at a CT of 600 °C. As can be seen, a large number of nano-sized precipitates were distributed in the matrix and displayed different morphologies according to Figure 9a. In Figure 9b, one of the particles was selected to further analyze the composition and structure. The selected area electron diffraction (SAED) used in conjunction with the EDS showed that the precipitates were rich in Fe with small amounts of Nb, Ti, C, and N in Figure 9c,d. As most of the Fe content came from the matrix, the precipitates were identified as (Nb, Ti) (C, N) and had an orientation relationship in ferrite of $[011]_{(Nb, Ti)(C, N)}/[011]_{\alpha\text{-Fe}}$.

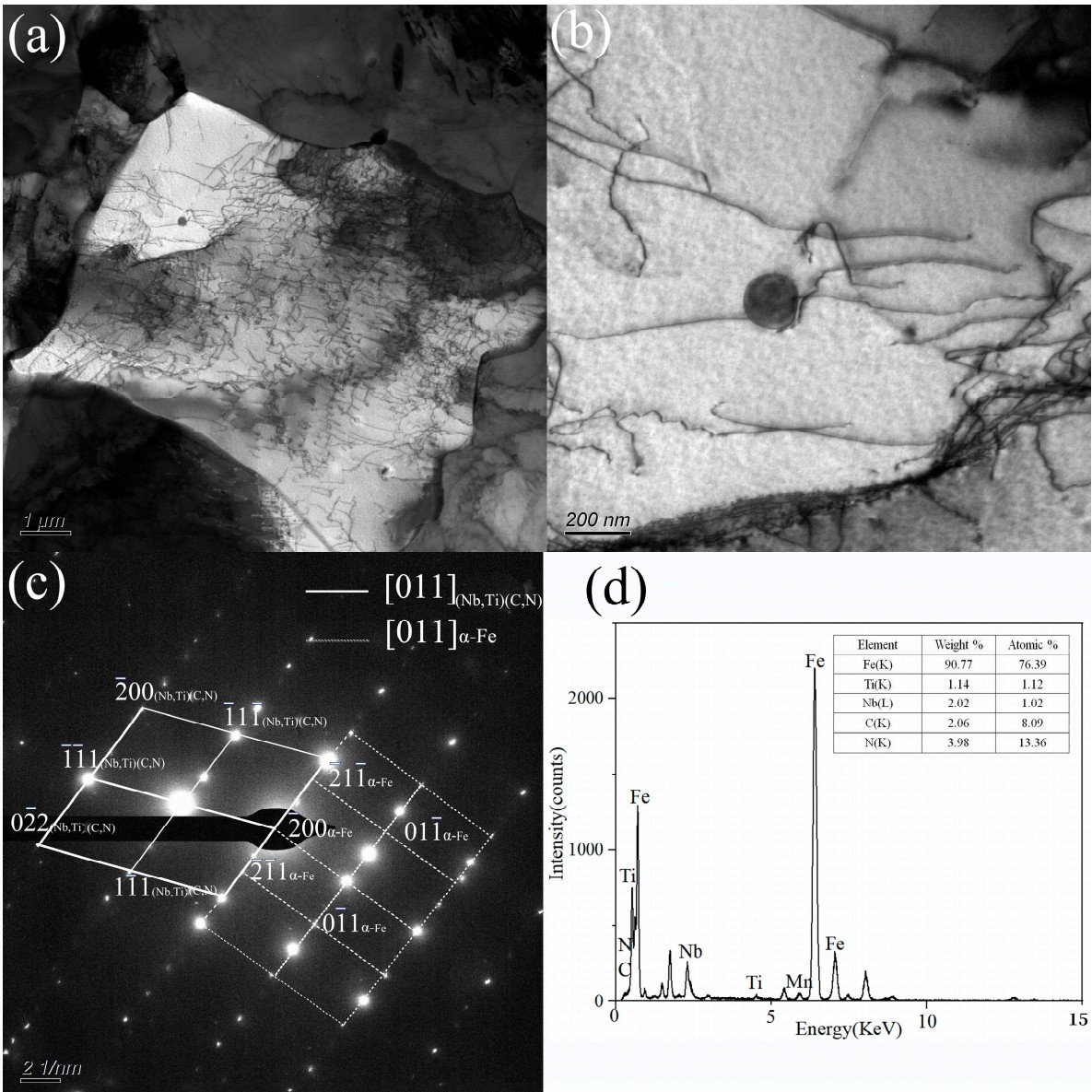

**Figure 9.** The bright field TEM thin foil (**a**) showing the morphology of precipitates and (**b**) at high magnification; (**c**) the SAED diffraction pattern; and (**d**) the EDS spectra of precipitates.

Figure 10a shows the elliptical morphology of the precipitates with a diameter of approximately 113.4 nm. The SAED and EDS analyses in Figure 10c,d show that the precipitates mainly contained Ti, C, and a small amount of sulfur. Part of the C and all the Cu were from the carbon film and the copper mesh, respectively. Therefore, the content of C in the precipitates could not be accurately measured. The precipitate had a face-centered

cubic structure with a lattice constant of 0.4317 nm. Therefore, it was deduced that the precipitate was TiC. The morphology of another precipitated phase in Figure 10d was round with a diameter of 132.2 nm. The EDS analysis showed that this precipitate comprised Ti, Nb, and C, as shown in Figure 10e. Combined with SAED analysis (Figure 10f), it was deduced that this was (Nb, Ti)C with a face-centered cubic structure and a lattice constant of 0.4427 nm.

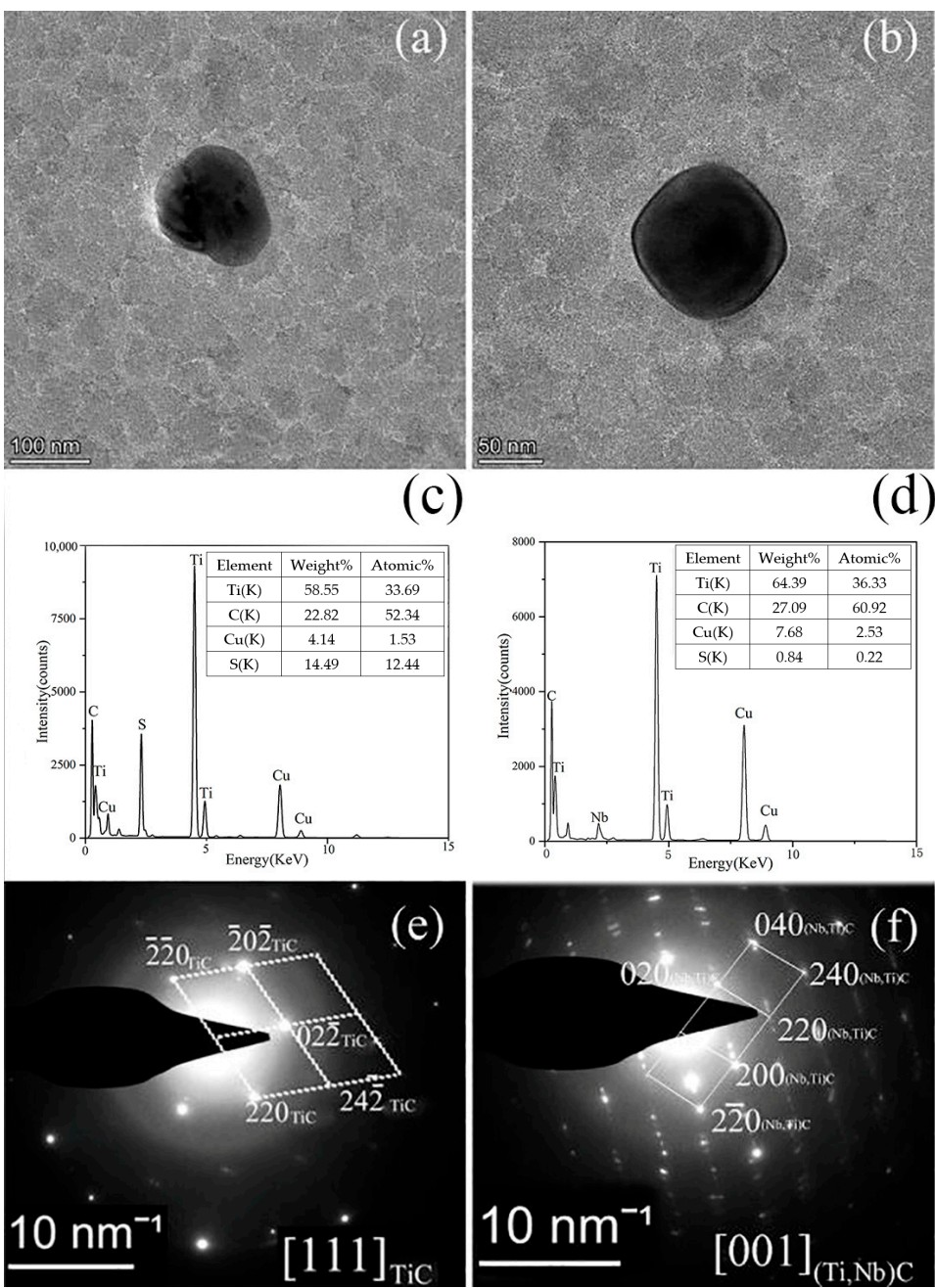

**Figure 10.** (**a**,**b**) TEM carbon replica micrographs; (**c**,**d**) EDS analysis; and (**e**,**f**) corresponding SAED patterns at CTs of 600 and 625 °C, respectively.

Figure 11 shows the morphology and size distribution of the precipitates at CTs of 600, 625, and 650 °C, respectively. A large number of nanoparticles were analyzed to ensure the accuracy of the analysis. It can be seen that the size of nanoparticles mainly ranged from 60 to 120 nm, and the nanoparticles smaller than 100 nm constituted more than 70%. The nano-sized carbide showed excellent thermal stability and maintained a small size after the rolling and coiling processes. The average diameters of these precipitates were found to be

85.6, 89.6, and 98.9 nm at CTs of 600, 625, and 650 °C, respectively. The volume fraction of the precipitates was calculated as 0.166%, 0.151%, and 0.156% at CTs of 600, 625, and 650 °C, respectively.

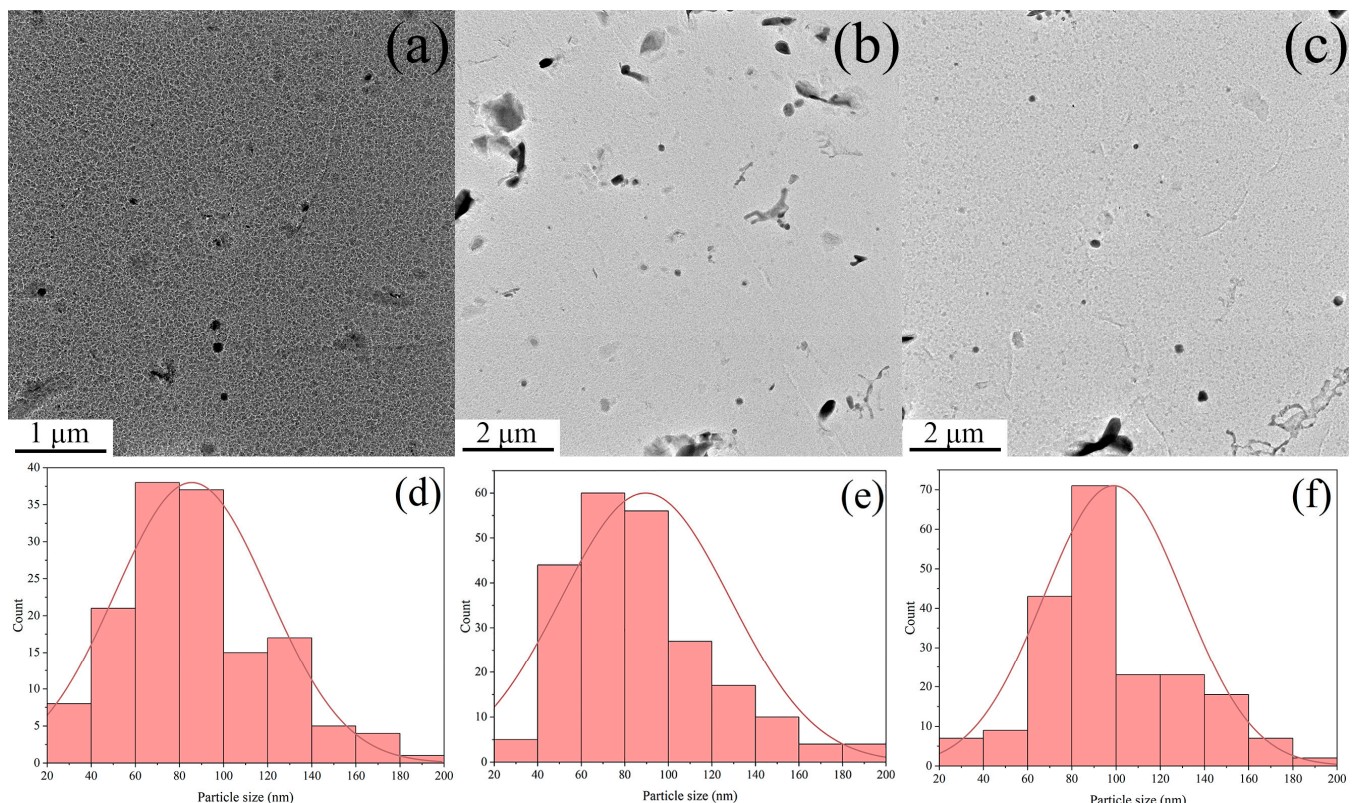

**Figure 11.** Carbon replica micrographs and particle size distribution at CTs of (**a**,**d**) 600; (**b**,**e**) 625; and (**c**,**f**) 650 °C.

## 4. Discussion

### 4.1. Solid Solution and Volume Fraction of the Precipitated Particles

The solubility of carbide and nitride-forming elements in austenite is dependent on temperature [35]. During the cooling process, the supersaturation of solute elements increases with the decrease in temperature. $Ti_xNb_yC$ can be used to represent (Nb, Ti)C, where x + y = 1. The solid solution formula of TiC and NbC can be expressed as follows in ferrite [36]:

$$\lg\{[Ti] \cdot [C]\}_\alpha = 4.4 - 9575/T, \tag{1}$$

$$\lg\{[Nb] \cdot [C]\}_\alpha = 3.9 - 9930/T. \tag{2}$$

where $\alpha$ is the concentration product in ferrite, and the ideal chemical ratios of TiC and NbC are 3.99 and 7.735, respectively. Thus [37]:

$$\frac{Ti - [Ti]}{\{C - [C]\}x} = 3.991, \tag{3}$$

$$\frac{Nb - [Nb]}{\{C - [C]\}y} = 7.735. \tag{4}$$

The volume fraction of precipitates can be expressed as follows [37]:

$$f_V = \left(\sum M_i - \sum [M_i] + C - [C]\right)\frac{\rho_{Fe}}{100\rho_{MC}}, \tag{5}$$

$$\rho_{MC} = x\rho_{M1C} + y\rho_{M2C}, \tag{6}$$

where $[M_i]$, $(M_i = \text{Ti, Nb})$, is the solid solution quantity of element M in ferrite, $[C]$ is the solid solution quantity of carbon in ferrite, $f_v$ is the volume fraction of precipitation particle in the matrix, $\rho_{Fe}$ is the density of TMCP Nb-Ti HSLA pipeline steel, $\rho_{TiC}$ and $\rho_{NbC}$ is the precipitated particle density in ferrite of TiC and NbC, respectively, and $\rho_{MC}$ is the total density of $\rho_{TiC}$ plus $\rho_{NbC}$ in ferrite. It should be noted that in order to make the model simpler, the N element in the matrix is considered to be completely transformed into TiN, and the content of the Ti element in the steel is modified according to the content of TiN.

Figure 12a shows the effect of temperature on the chemical formula coefficient of the $\text{Ti}_x\text{Nb}_y\text{C}$ in micro-alloyed steel. It can be seen that the $\text{Ti}_x\text{Nb}_y\text{C}$ is rich in a higher proportion of Ti atoms from 500 to 800 °C. As the temperature decreases, the proportion of Ti atoms decreases slightly. Figure 12b shows the effect of temperature on the equilibrium solid solution content of Ti, Nb, and C elements. The equilibrium solid solution amounts of Ti, Nb, and C elements gradually decrease with temperature and the downward trend gradually becomes slower. This shows that the $\text{Ti}_x\text{Nb}_y\text{C}$ is more prevalent at high temperatures. Figure 12c,d show the temperature dependence of the $\text{Ti}_x\text{Nb}_y\text{C}$ in ferrite. The volume fraction of the $\text{Ti}_x\text{Nb}_y\text{C}$ increases with a decrease in temperature, and when the temperature drops down to 950 °C, the increase in the volume fraction of the precipitates gradually becomes slower.

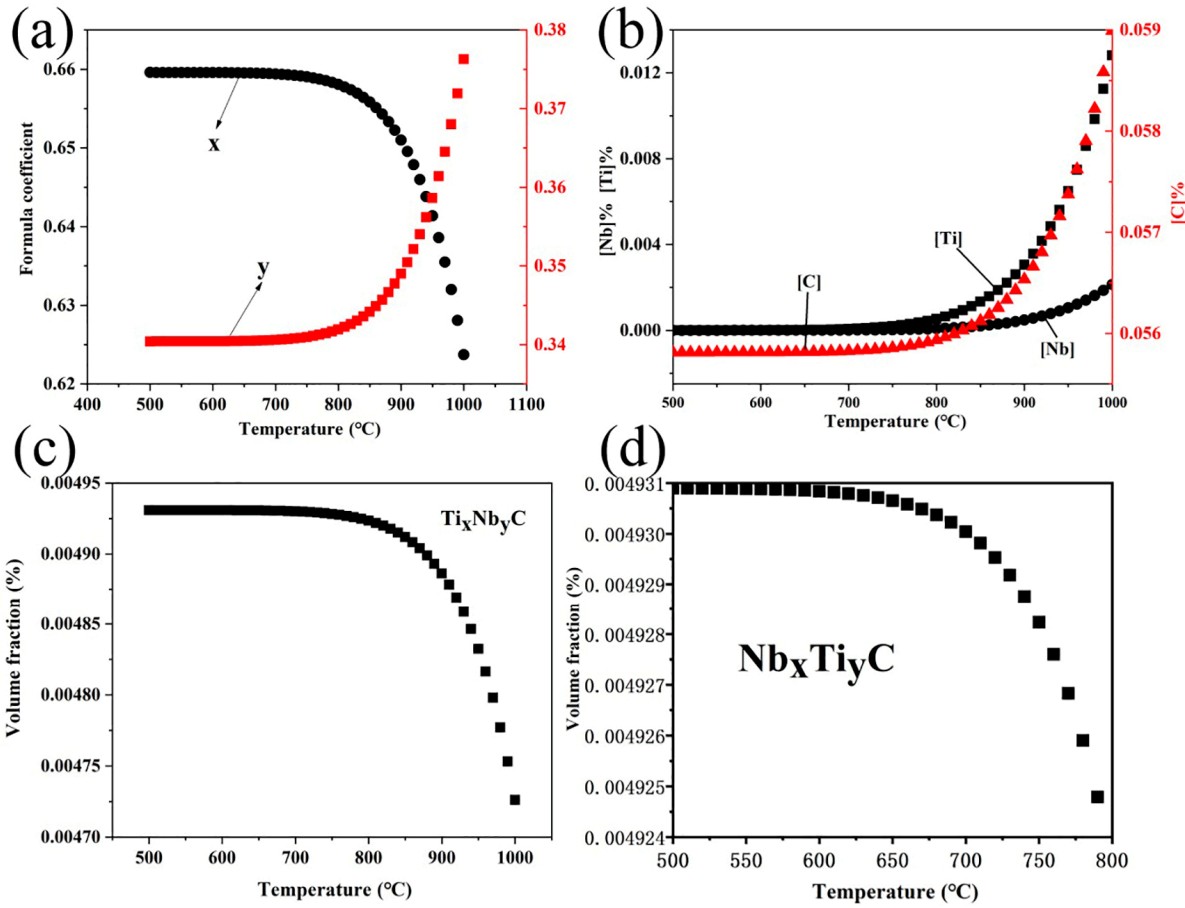

**Figure 12.** The effects of temperature on (**a**) the formula coefficient, (**b**) the equilibrium solution of Ti, Nb, and C, and (**c,d**) the precipitation volume fraction.

### 4.2. Nucleation of Precipitate during Strain-Induced Precipitation

Figure 9 shows bright field TEM micrographs showing the heterogeneous nucleation of (Nb,Ti)C nanoparticles on dislocations. These strain-induced precipitates on dislocations play a major role as a strengthening mechanism in TMCP Nb-Ti HSLA pipeline steel. The precipitation occurs at different stages of the hot-rolling process. According to Figure 9, the

rolling process introduces lattice defects such as dislocations. The place where dislocations exist can serve as a channel for rapid solute diffusion, and then promote the formation of a rich solute core. As a result, the precipitates preferentially occur at the austenite grain boundary or dislocation position [38].

A model of strain-induced precipitation is proposed by assuming that the precipitates nucleate along dislocation. During the nucleation process, the change in free energy and critical radius due to the formation of a spherical nucleus can be expressed as follows [36,39]:

$$
\begin{aligned}
\Delta G &= \Delta G_C + \Delta G_i + \Delta G_d \\
&= \frac{4}{3}\pi r^3 \Delta G_V + 4\pi r^2 \sigma - 2Ar
\end{aligned}
\tag{7}
$$

where $\Delta G_c$ refers to chemistry free energy, $\Delta G_i$ refers to interfacial free energy, $\Delta G_d$ is dislocation core energy per unit, $r$ is the radius of the precipitate, $\Delta G_V$ is the volume free energy, $A$ is the core energy of dislocation line per unit, equal to $\frac{Gb^2}{4\pi(1-v_b)}$ for edge dislocation or $\frac{Gb^2}{4\pi}$ screw dislocation, and $\sigma$ is the interfacial energy between precipitate and austenite. The volume free energy ($\Delta G_v$) can be obtained as follows [40]:

$$
\Delta G_v = \frac{1}{V_m}\{-19.1446B_1 + 19.1446T\{A_1 - \log\{[\mathrm{M}]_0[\mathrm{C}]_0\}\}\},
\tag{8}
$$

where $V_m$ refers to the molar volume of MC precipitates, $[\mathrm{M}]_0$ and $[\mathrm{C}]_0$ refer to M and C atoms in solution. During the nucleation process, the change in critical radius due to the formation of a spherical nucleus can be expressed as follows [36]:

$$
r_0 = -\frac{\sigma}{\Delta G_V}\left[1 + \left(1 + \frac{A\Delta G_V}{2\pi\sigma^2}\right)^{\frac{1}{2}}\right].
\tag{9}
$$

The critical radius can be expressed as [37]:

$$
r^* = -\frac{\sigma}{\Delta G_V}\left[1 - \left(1 + \frac{A\Delta G_V}{2\pi\sigma^2}\right)^{\frac{1}{2}}\right].
\tag{10}
$$

The critical nucleation energy can be obtained as [37]:

$$
\Delta G^* = \frac{16\pi\sigma^3}{3\Delta G_V^2}\left(1 + \frac{A\Delta G_V}{2\pi\sigma^2}\right)^{\frac{3}{2}}.
\tag{11}
$$

The undetermined nucleation rate of precipitates can be expressed by the classical nucleation theory [26]:

$$
\begin{aligned}
I &= n_V a^* pv \cdot \pi\rho b^2 \exp(-\frac{Q_d}{kT})\exp(-\frac{\Delta G^*}{kT}) \\
&= K \cdot \rho b^2 \cdot d^{*2} \exp(-\frac{Q_d}{kT})\exp(-\frac{\Delta G^*}{kT})
\end{aligned}
\tag{12}
$$

where $n_v$ is the number of nucleation sites per unit volume, $a^*$ is the number of atoms on the surface of a single critical nucleate, $d^*$ refers to the size of the critical nucleus, $p$ is the probability of atom jumping, $v$ is the frequency of atom hopping in the matrix, $\rho$ is the dislocation density, $b$ is the burgers vector, $Q_d$ is the diffusion activation energy along dislocation, $K$ is the temperature-dependent parameter, $k$ is the Boltzmann constant, and $T$ is the temperature.

### 4.3. Growth of Precipitate during Strain-Induced Precipitation

The diffusion rate of Ti, Nb, and C atoms along the dislocation line is faster than through the matrix, so we supposed that the diffusion of solute atoms along the dislocation line controls the growth process after precipitation. The diffusion of Ti, Nb, and C

atoms along the dislocation line is a two-dimensional process that can be described by the following equation [41]:

$$y = \lambda (Dt)^{\frac{1}{2}}, \tag{13}$$

where $y$ refers to the diameter of the dislocation line, $D$ refers to the diffusion coefficient of the element, $t$ refers to time, and $\lambda$ refers to a constant related to the supersaturation of Ti, Nb, and C atoms. $\lambda$ can be expressed by the following equation [36]:

$$\lambda = 2 \left[ \frac{2(C_0 - C_l)}{C_p - C_l} \right], \tag{14}$$

where $C_0$ refers to the initial concentration of Ti, Nb, and C atoms, $C_l$ refers to the concentration of Ti, Nb, and C atoms at the interface between precipitate and matrix, and $C_p$ refers to the concentration of Ti, Nb, and C atoms in the precipitate.

### 4.4. Precipitation–Time–Temperature (PTT) Curves

The precipitation–time–temperature (PPT) curves for the TiC and $Ti_xNb_yC$ can be calculated as follows [37,42]:

$$\lg \frac{t_{0.05}}{t_0} = -1.28994 - 2\lg d^* + \frac{1}{\ln 10} \frac{(1 + \beta)^{\frac{3}{2}} \Delta G^* + \frac{5}{3} Q}{kT}, \tag{15}$$

where $t_0$ is a temperature-independent function, $t_{0.05}$ is the start time when 5% of the precipitated phase precipitates, $d^*$ is the critical nucleation size, $\Delta G_v$ is the volume free energy, $\Delta G^*$ is a critical nuclear energy, $Q$ is the activation energy of atoms, $k$ is the Boltzmann constant, and $T$ is the temperature.

According to the formula above, it can be inferred that high dislocation density promotes the nucleation of precipitates along the dislocation line, such as $Ti_xNb_yC$ nanoparticles. A large number of fine (Nb, Ti)C nanoparticles are formed due to the relatively high dislocation density when the Nb-Ti HSLA pipeline steel is coiled at 600 °C instead of 650 °C.

The calculated Gibbs free energy, critical nucleation size, activation energy, and PTT curves are shown in Figure 13 for $Ti_xNb_yC$ and TiC precipitates. It can be seen from Figure 13a that the Gibbs free energy of $Ti_xNb_yC$ is larger than that of TiC for the same temperature because $Ti_xNb_yC$ has higher supersaturation. As a result, the calculated $Ti_xNb_yC$ critical embryos are finer than those of the TiC, as shown in Figure 13b,c.

In Figure 13d, it is obvious that the PTT curves calculated present a typical "C" shape. It can be seen that $Ti_xNb_yC$ precipitates have a greater driving force for nucleation than that of TiC. From this theoretical model, it is apparent that niobium precipitates first and thereafter Ti atoms gradually replace Nb atoms to form niobium and titanium compounds, and finally titanium compounds in TMCP Nb-Ti HSLA pipeline steel. In other words, $Ti_xNb_yC$ and (Nb, Ti)(C, N) preferentially form before TiC nanoparticles during coiling.

### 4.5. Estimation of Precipitation Strengthening

The yield strength of steel can be affected by a combination of factors such as dislocation strengthening, solution strengthening, grain refinement strengthening, and precipitation strengthening. As for Nb-Ti HSLA steel, the TiC, (Nb, Ti)C, and (Nb, Ti)(C, N) precipitates formed in the matrix can provide extra strength through precipitation strengthening. The incremental strength ($\sigma_p$) of the steel at different CTs can be calculated through the Ashby–Orowan formula (Formula (16)) [42], which is considered a good choice for calculating the effect of precipitation strengthening through a relationship between strength, diameter, and content of precipitates:

$$\sigma_p = \frac{0.538 G b \sqrt{f}}{X} \ln \frac{X}{2b}, \tag{16}$$

where $\sigma_p$ is the incremental strength, MPa; $G$ is the shear modulus, 81,600 MPa for Fe matrix; $b$ is the burgers vector, 0.248 nm for body-centered cubic (BCC); $X$ is the average diameter of the particles, nm; and $f$ is the volume fraction of the precipitates. Based on the statistical analysis of precipitates in Figure 11, $\sigma_p$ can be calculated as 26.67, 24.54, and 23.03 MPa at the CTs of 600, 625, and 650 °C, respectively. The average diameter and volume fraction of the precipitates and $\sigma_p$ at different CTs are shown in Figure 14. It can be concluded that the average diameter of the precipitates increased with the CT while the volume fraction showed no significant change. According to Formula (16), the small size and large volume fraction are beneficial for strength, so the $\sigma_p$ value decreased with the CT.

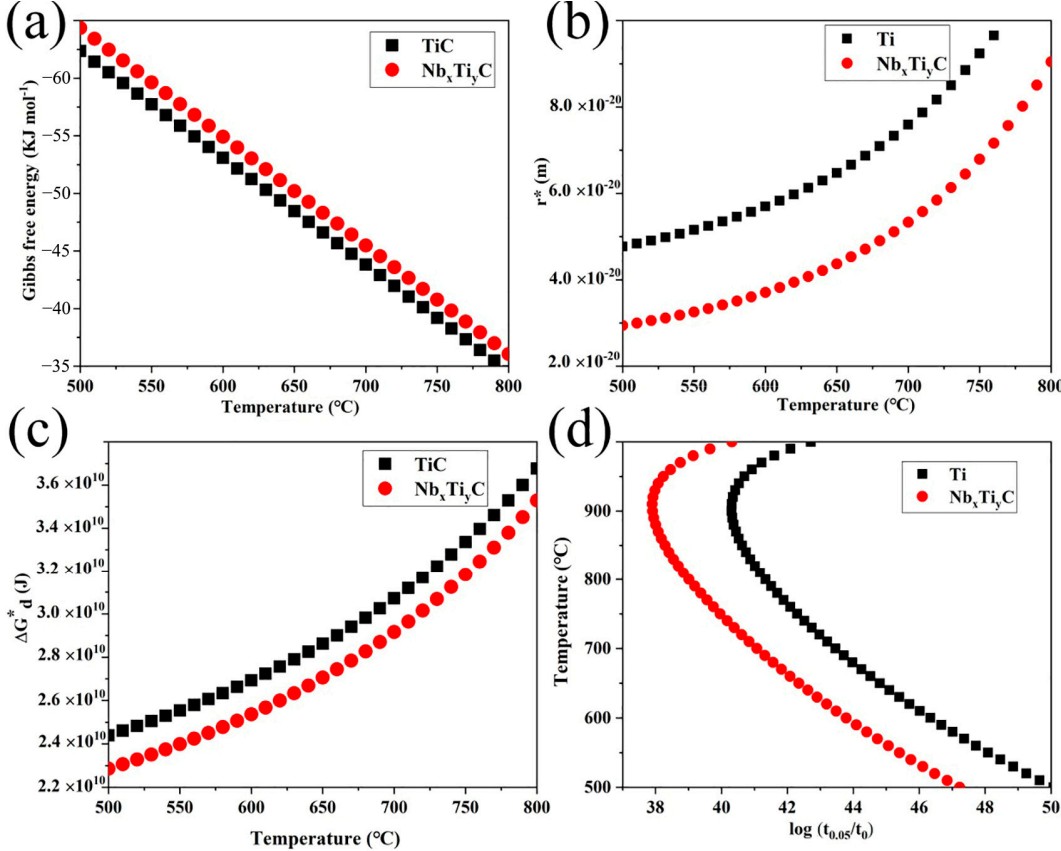

**Figure 13.** (**a**) Gibbs free energy, (**b**) critical nucleation size, (**c**) critical nucleation energy, and (**d**) PTT curves of TiC and $Ti_xNb_yC$ precipitates based on the thermodynamic model in TMCP Nb-Ti HSLA pipeline steel.

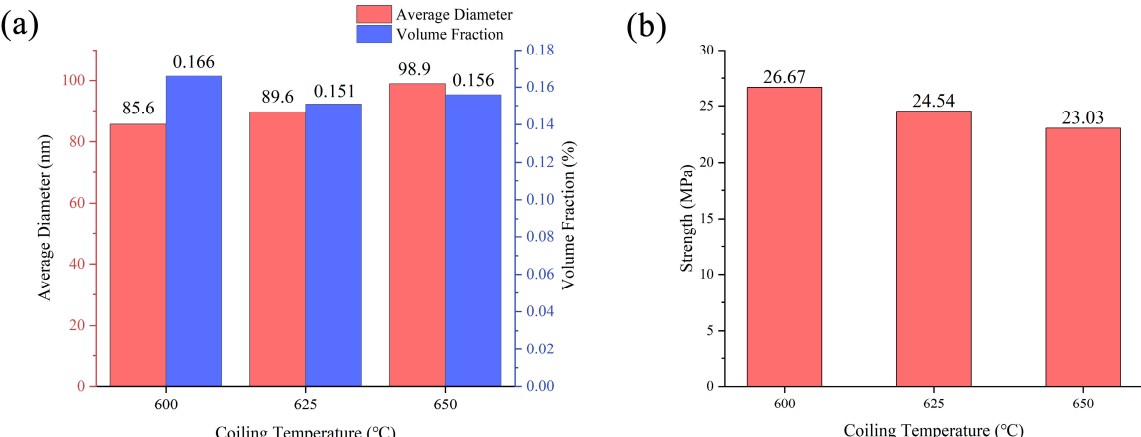

**Figure 14.** (**a**) The average diameter and volume fraction values of the precipitates and (**b**) $\sigma_p$ at CTs of 600, 625, and 650 °C.

## 5. Conclusions

In this study, the microstructure and precipitates of Nb-Ti HSLA pipeline steel with 72% heavy reduction after six passes of a hot-rolled process were studied at different coiling temperatures ranging between 600 °C to 650 °C. The main conclusions are summarized as follows:

1. The microstructures consisted of acicular ferrite with high-density dislocation, polygonal ferrite, and pearlite. The volume fractions of pearlite were found to be 11.48, 11.68, and 12.37%, while the volume fractions of the acicular ferrite were 31.48, 29.65, and 23.28% at coiling temperatures of 600, 625, and 650 °C, respectively. The pearlite increased with the coiling temperature while the acicular ferrite decreased with the coiling temperature;

2. The TiC, (Nb,Ti)C, and (Nb,Ti)(C,N) precipitates formed in the matrix, and the orientation relationship with the matrix was $[011]_{(Nb,Ti)(C,N)}//[011]_{\alpha\text{-Fe}}$. The average size of precipitates was 96.0, 98.9, and 105.6 nm, while the volume fraction was 0.166%, 0.151%, and 0.156% at coiling temperatures of 600, 625, and 650 °C, respectively. The estimation of incremental yield strength through precipitation strengthening could reach 26.67 MPa at a coiling temperature of 600 °C and decrease with the coiling temperature;

3. According to a theoretical model based on classical nucleation and growth theory, (Nb, Ti)C and (Nb, Ti)(C, N) exhibit a high driving force and, therefore, can form before TiC precipitates.

**Author Contributions:** Conceptualization, methodology, and writing—original draft preparation, Y.L. and W.Y.; software, W.Y.; and writing—review and editing, Z.T. and C.W.S. All authors have read and agreed to the published version of the manuscript.

**Funding:** This work was financially supported by Urgently Needed Talent Projects in Key Supported Regions of Development and Reform Commission of Shandong Province; China and South Africa's Science and Technology Innovation Cooperation Project (Grant Number 2017YFE0113400); and the fifth batch of projects in the Panxi Test Zone (No. 2022P5C4K03).

**Data Availability Statement:** The raw data supporting the conclusions of this article will be made available by the authors on request.

**Conflicts of Interest:** The authors declare no conflicts of interest.

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
