# Peer review of "Effect of Coiling Temperature on Microstructures and Precipitates in High-Strength Low-Alloy Pipeline Steel after Heavy Reduction during a Six-Pass Rolling Thermo-Mechanical Controlled Process"

_metals, doi:10.3390/met14020249_

Round 1

Reviewer 1 Report

Comments and Suggestions for Authors

The paper deals with the processing and microstructural analysis of an HSLA steel at different coiling temperatures, and therefore the paper fits with the purpose of the “Metals” journal.

The paper is written in proper English, and the structure follows the standards of scientific papers, being the abstract enough to gather the idea of the results presented.  The extension of the paper is enough. The figures have good quality and are necessary for understanding the text.

However, just some items should be taken into account and amended:

In the title is said "simulated" and perhaps this word could create confusion, as in a first instance, as it reminds of numerical calulus and could be misleading.  I suppose the authors refer to the process carried out in a laboratory machine (the Geeble) and not in the actual rolling line.  Later on also appears simulator when referring to the machine, consider revise also these.  In fact, in the Geeble page, the machine is defined as "The Gleeble 3500 is a fully integrated digital closed loop control thermal and mechanical testing system"

Figure 1b) revise CT temperatures (do not match logically the plot)

Page 6, line 149. Shouldn't it be the opposite trend?

 "Therefore, the volume fraction of the AF was calculated from the EBSD data and was found to be 31.48, 29.65 and 23.28% for Ct of 600, 625 and 650 ℃, respectively. In other  words, the volume fraction of the AF gradually increased with the CT"  should read "decreased with the CT"

Page 7,  line 159 "circlar" revise if it is correct (->circular?)

No other issues have been detected.

Comments on the Quality of English Language

The english is fine enough, just few expressions that could be improved by an English native.

Reviewer 2 Report

Comments and Suggestions for Authors

The reviewer congratulates the extensive and well-prepared research material. The photographs of the structure are well selected and of good quality. However, the following issues need to be supplemented and corrected in the article:

1.Describe the research purpose in detail (in Line 61-68). Why 6-pass rolling and not others? E.g. 4-pass rolling, same temperature range.

2. The utilitarian purpose was formulated, line 67-68. There is nothing in the conclusions on this subject (complete)

3. The reviewer suggests that in the introduction the authors comment on the results of the cited studies of other researchers in the context of their own research.

What is missing in the literature that they undertake their research.

What scientific novelty do the authors declare?

4. The conclusions should not be just a repetition of the obtained results, but a broader general commentary on the results about the observed phenomena, regularities, etc. In the introduction, the authors declared
technological recommendations, but this is not included in the conclusions.

Reviewer 3 Report

Comments and Suggestions for Authors

The reviewed paper titled “Effect of coiling temperature on microstructures and precipitates in high strength low alloy pipeline steel after simulated heavy reduction during six-pass-rolling thermo-mechanical controlled process” is an interesting work associated with structure valuation during hot plastic forming.

The authors did not avoid minor errors and inaccuracies. Notes for reflection and minor correction of the work:

Keywords -  please added “steel”; word “multi-pass”- remove

Introduction

- citation order not followed, for example Number [16] is missed  (line 44 -45)

- something more could be written about the practical use of this type of steel in industry

Materials and Methods

No information about the dimensions of the ingot. Was an analysis of the chemical composition performed along the length and width of the ingot? From which part of the ingot were the samples taken? In cast material, there are large differences in the material properties and its susceptibility to deformation. In my opinion, the ingot should have been initially deformed, samples cut out and then homogenized.

Result

-  The quality of the work is reduced by the insufficient quality of the figures, some are even illegible, e.g. figs. 2, 9d, 12, 13, 14

Conclusions

One thing is missing about the conclusion regarding what generally results from the research and what impact it will have, e.g. in industry

Comments on the Quality of English Language

English language correct
